# Identification of Key Active Constituents in *Eucommia ulmoides* Oliv. Leaves Against Parkinson’s Disease and the Alleviative Effects via 4E-BP1 Up-Regulation

**DOI:** 10.3390/ijms26062762

**Published:** 2025-03-19

**Authors:** Yuqing Li, Ruidie Shi, Lijie Xia, Xuanming Zhang, Pengyu Zhang, Siyuan Liu, Kechun Liu, Attila Sik, Rostyslav Stoika, Meng Jin

**Affiliations:** 1Biology Institute, Qilu University of Technology (Shandong Academy of Sciences), 28789 East Jingshi Road, Jinan 250103, China; 2Engineering Research Center of Zebrafish Models for Human Diseases and Drug Screening of Shandong Province, 28789 East Jingshi Road, Jinan 250103, China; 3Shandong Provincial Key Laboratory of Molecular Engineering, School of Chemistry and Chemical Engineering, Qilu University of Technology (Shandong Academy of Sciences), Jinan 250353, China; 4University Research and Innovation Center, Obuda University, Bécsi út 96B, H-1034 Budapest, Hungary; 5Institute of Physiology, Medical School, University of Pecs, H-7624 Pecs, Hungary; 6Institute of Clinical Sciences, Medical School, University of Birmingham, Birmingham B15 2TT, UK; 7Department of Regulation of Cell Proliferation and Apoptosis, Institute of Cell Biology, National Academy of Sciences of Ukraine, 79005 Lviv, Ukraine

**Keywords:** PD, RPPA analysis, 4E-binding protein 1, zebrafish, SH-SY5Y

## Abstract

Parkinson’s disease (PD) is the second most common progressive neurodegenerative disorder, affecting an increasing number of older adults. Despite extensive research, a definitive cure remains elusive. *Eucommia ulmoides* Oliv. leaves (EUOL) have been reported to exhibit protective effects on neurodegenerative diseases, however, their efficacy, key active constituents, and pharmacological mechanisms are not yet understood. This study aims to explore the optimal constituents of EUOL regarding anti-PD activity and its underlying mechanisms. Using a zebrafish PD model, we found that the 30% ethanol fraction extract (EF) of EUOL significantly relieved MPTP-induced locomotor impairments, increased the length of dopaminergic neurons, inhibited the loss of neuronal vasculature, and regulated the misexpression of autophagy-related genes (*α-syn*, *lc3b*, *p62*, and *atg7*). Assays of key regulators involved in PD further verified the potential of the 30% EF against PD in the cellular PD model. Reverse phase protein array (RPPA) analysis revealed that 30% EF exerted anti-PD activity by activating 4E-BP1, which was confirmed by Western blotting. Phytochemical analysis indicated that cryptochlorogenic acid, chlorogenic acid, asperuloside, caffeic acid, and asperulosidic acid are the main components of the 30% EF. Molecular docking and surface plasmon resonance (SPR) indicated that the main components of the 30% EF exhibited favorable binding interactions with 4E-BP1, further highlighting the roles of 4E-BP1 in this process. Accordingly, these components were observed to ameliorate PD-like behaviors in the zebrafish model. Overall, this study revealed that the 30% EF is the key active constituent of EUOL, which had considerable ameliorative effects on PD by up-regulating 4E-BP1. This suggests that EUOL could serve as a promising candidate for the development of novel functional foods aimed at supporting PD treatment.

## 1. Introductions

Parkinson’s disease, ranked as the second most frequent neurodegenerative condition, is experiencing a rising incidence in older adults [1,2]. Characteristic features of PD include the depletion of dopamine-producing neurons in the substantia nigra and the buildup of α-synuclein (α-syn) aggregates [3]. The specific loss of dopaminergic neurons within the substantia nigra, accompanied by the presence of α-syn aggregates in Lewy bodies, leads to decreased dopamine levels, abnormal brain activity, and impaired movements, which are among the most prominent symptoms of PD [4,5]. The pathological and physiological mechanisms of PD are intricate, including endoplasmic reticulum dysfunction, cytoskeletal disruption, impaired microtubule, autophagy and mitochondrial dysfunction [6,7,8]. At present, the primary therapeutic strategy for PD is the administration of medications, such as levodopa and dopamine agonists [9,10,11]. For instance, rasagiline is a selective and irreversible MAO-B inhibitor that prevents the breakdown of dopamine in the brain, leading to increased dopamine levels and subsequent relief of PD patients’ symptoms [12,13]. As a precursor to dopamine, levodopa can penetrate the blood–brain barrier, and is then transformed into dopamine within the cerebral cortex, restoring the diminished dopamine concentrations that result from the depletion of dopaminergic cells, thus mitigating the symptoms of PD [14,15,16]. While pharmacological interventions have postponed the progression of PD, current treatment methods primarily focus on symptom control rather than altering the underlying course of the disease [17,18,19]. In addition, their long-term usage causes side effects and drug resistance [20,21,22]. Therefore, there is a significant need to identify safe and effective anti-PD constituents. Recently, there has been a growing focus on natural products due to their promising role in the advancement of treatments for PD. It has been demonstrated that extracts from *Calendula officinalis L*. exhibit a protective effect against PD-like pathology in zebrafish by promoting the activation of autophagy [23]. *Sanghuangprous vaninii* extracts have been found to attenuate the loss of dopaminergic neurons and alleviate PD-like symptoms in zebrafish models induced by 1-methyl-4-phenyl-1,2,3,6-tetrahydropyridine (MPTP) [24]. In addition, morroniside, an active component in the *Cornus officinalis* Sieb. Et Zucc, activates the Nrf2/ARE signaling pathway in the MPTP-induced mouse PD model, enhancing antioxidant capacity and preventing aberrant lipid metabolism, thereby shielding dopaminergic neurons from ferroptosis [25].

*Eucommia ulmoides* Oliv. (www.theplantlist.org), named by the British botanist Daniel Oliver and known as “Duzhong” in Chinese, is widely used [26]. *Eucommia ulmoides* Oliv. leaves (EUOL) can be used to treat dizziness, blurred vision, soreness, weakness in the waist and knees, and insufficiency of the liver and kidney. Traditionally, only the barks have been employed for medicinal use, whereas the leaves have generally been underutilized, despite containing nearly the same bioactive components as the barks [27]. The active components in EUOL mainly include polyphenols, sterols, terpenoids, flavonoids, polysaccharides, and phenylpropanoids [28]. EUOL, embodying the concept of ‘food as medicine’, have emerged as a promising novel functional food due to their rich content of bioactive compounds, which offer a wide range of health benefits including pain relief, calming, lowering blood pressure, antidepressant properties, controlling cholesterol metabolism, and reducing hyperlipidemia [29,30,31,32]. Importantly, it has been reported that EUOL and its active components have protective effects on Alzheimer’s disease, which is the neurodegenerative disorder with the highest incidence [33]. In our previous study, we found that extracts of EUOL have therapeutic effects on PD [34]. However, the information provided by the previous research fails to provide clear evidence for the key active constituents and underlying mechanisms of EUOL in PD treatment.

The zebrafish PD model exhibits similar pathological features to PD patients, including the occurrence of Lewy bodies, the gradual depletion of dopaminergic neurons in the brain, and impaired motor function [35,36]. In addition, with approximately 70% of human genes having a zebrafish orthologue, the genetic similarity supports the relevance of findings in this model [37]. Further, the widespread usage of the zebrafish PD model in drug screening and mechanism exploration underscores the recognition and reliability of zebrafish as a valuable model organism in PD studies [38,39]. MPTP is a lipid-soluble neurotoxin that chemically ablates nigral dopaminergic neurons [40]. MPTP could cross the blood–brain barrier and be metabolized to 1-methyl-4-phenylpyridinium (MPP^+^), which has a high affinity for dopamine transporter and eventually damages neurons by inhibiting mitochondrial function [41,42]. MPTP could cause loss of dopaminergic neurons in the ventral diencephalon and decreased locomotor activity with corresponding PD-like symptoms in zebrafish [43,44,45]. As a natural hydroxylated dopamine analog, 6-hydroxydopamine (6-OHDA) could enter the cells via the dopamine transporter, causing severe cellular damage, including mitochondrial dysfunction and apoptosis [46,47]. The neuroblastoma SH-SY5Y cell line closely resembled in vivo dopaminergic neurons was responsive to 6-OHDA, and is widely used in PD research [48]. Our previous study used 6-OHDA-induced neurotoxicity in SH-SY5Y cells to explore the protective effect of berberine against PD [49].

RPPA technology is an extremely sensitive, precise, and highly efficient method for multiplex analysis to identify biomarker candidates and define targets of the pathogenesis [50,51]. When comparing the protein expression data of biomarker studies between ELISA and RPPA, a high degree of consistency and correlation was observed, emphasizing the repeatability and clinical compatibility of RPPA [52]. RPPA technology is commonly used to study tumor-related diseases, but its application in neurodegenerative diseases is still limited. Our previous study showed that RPPA technology offers a new approach to PD diagnosis and treatment [53]. The eukaryotic translation initiation factor 4E-binding protein 1 (4E-BP1) belongs to a group of translation inhibitory proteins. Research suggests that increasing 4E-BP1 expression can induce the mitochondrial unfolded protein response, and may be an effective strategy for treating several neurodegenerative diseases, particularly PD [54]. It has been reported that activating 4E-BP1 could prevent dopaminergic neurodegeneration in the *Drosophila* PD model [55].

The key active constituents and pharmacological mechanisms of EUOL against PD have not been deciphered. Thus, we hypothesized that the optimal active constituents of EUOL alleviated PD symptoms by up-regulating 4E-BP1, thereby restoring translational homeostasis and mitigating dopaminergic neuron loss. To test this hypothesis, we employed the MPTP-induced zebrafish PD model to efficiently identify the optimal active constituents from EUOL against PD, and subsequently assayed the key regulators involved in PD to verify their potential in multi-models. The underlying mechanisms were studied through RPPA technology and confirmed by Western blot, phytochemical analysis, molecular docking, and SPR (Figure 1).

## 2. Results

### 2.1. Neuroprotective Effect of Different Fractions of EUOL Extract on Zebrafish PD-like Behavior

It has been reported that the loss of dopaminergic neurons in the pathophysiology of PD disrupts the supply of dopamine to the striatum, impairing motor capacity [56,57]. To compare the effect of different fractions of EUOL extract on PD-like behavior in zebrafish, we assessed the total distance and velocity traveled by the zebrafish. In the locomotor activity test and the light/dark challenge test, the total distance and velocity traveled in the MPTP-treated group were significantly reduced compared to the Ctl group, which is consistent with the previous studies (Figure 2A,C) [49]. Co-treatment with rasagiline, TWE, WF, 10% EF, and 30% EF reversed the MPTP-induced decline in the total distance traveled by each zebrafish and the corresponding average swimming speed (Figure 2A,B). In the light/dark challenge test, an apparent recovery in response to stimuli, as well as a significant increase in the total distance and velocity traveled, were observed in the 10% EF and 30% EF-treated group compared to the MPTP-treated group (Figure 2C,D), among which 30% EF showed a similar effect to that of the rasagiline group. In summary, the 30% EF effectively alleviated locomotor impairments in zebrafish, showing the optimal potential compared to other fractions.

### 2.2. Effect of Different Fractions of EUOL Extract on PD Neuronal and Vascular Phenotype in Zebrafish PD Model

We evaluated the dopaminergic neurons to assay the neuroprotective effect of different fractions of EUOL extract on the MPTP-induced PD zebrafish model. Zebrafish treated with MPTP induced an apparent decrease in the dopaminergic neuron length (Figure 3A,B). However, co-treatment with 30% EF reversed this decrease to a normal level when the length of dopaminergic neurons was measured and compared with the Ctl group and rasagiline group (Figure 3B). The possible protective effect of different fractions of EUOL extract on MPTP-induced injured neuronal vasculature in the brain was also assessed. MPTP administration resulted in a significant loss of cerebral neuronal vasculature (Figure 3C). Co-treatment with rasagiline and 30% EF protected against the loss and disorganization of vasculature induced by MPTP (Figure 3C). Collectively, the 30% EF effectively ameliorated the damage to dopaminergic neurons and neuronal vasculature in zebrafish compared to other fractions.

### 2.3. Regulatory Effect of Different Fractions of EUOL Extract on Core PD-Related Genes

Autophagy is a critical process in the pathological mechanisms of PD, with *atg7*, *α-syn*, *lc3b*, and *p62* being widely recognized as core regulators in the PD pathogenesis [58,59]. We observed significantly up-regulated mRNA expression of *α-syn*, *lc3b*, and *p62* in the MPTP-administered group when compared to the Ctl group (Figure 4A,C,D). In contrast, rasagiline, WF, 10% EF, 30% EF, 50% EF, 70% EF, and 95% EF significantly reduced the expressions of *α-syn* (Figure 4A). Rasagiline, WF, 30% EF, 70% EF, and 95% EF also significantly reduced the expressions of *lc3b* (Figure 4C). There was a significant decline in the expression level of *p62* upon treatment with rasagiline, 30% EF, 70% EF, and 95% EF (Figure 4D). We found that the transcript level of *atg7* was significantly down-regulated in the MPTP group compared to the Ctl, while rasagiline, 30% EF, and 50% EF reversed this decrease (Figure 4B). By analyzing the expression of the genes mentioned above, we found that the 30% EF exhibited the best activity.

### 2.4. Verification of the Main Active Constituent Against PD by Assaying the Key Regulators Involved in PD

Considering the results presented above, we concluded that the 30% EF is the main active constituent of EUOL extract against PD. To further verify the anti-PD effect of 30% EF, we tested the expression of key autophagy regulators *α-syn* and LC3B. The location and expression of *α-syn* were determined using whole-mount in situ hybridization in the Ctl, rasagiline, TWE, and 30% EF (25 μg/mL, 50 μg/mL, 100 μg/mL) treated groups. Dense staining, indicative of *α-syn* expression, was observed in the zebrafish brains, particularly in the telencephalon and midbrain. Compared to the Ctl group, the expression of *α-syn* was reduced after treatment with rasagiline and 30% EF. 50 μg/mL of 30% EF showed the most significant effect (Figure 5A).

SH-SY5Y cells treated with 6-OHDA exhibited strong green fluorescence derived from LC3B-EGFP and red fluorescence emitted by RFP-Mito. The overlap between the green and red fluorescence indicated the accumulation of LC3B in mitochondria. Treatment with 6-OHDA led to intense green fluorescence from EGFP-LC3B, suggesting that 6-OHDA inhibited autophagosome degradation, causing an accumulation of LC3B protein. The co-treatment of cells with 6-OHDA and 30% EF significantly reduced the green fluorescence, which suggested that 30% EF could activate autophagy, and thereby down-regulate the accumulation of LC3B in 6-OHDA damaged SH-SY5Y cells (Figure 5B). These results indicated that the 50 μg/mL of 30% EF exhibited neuroprotective activity through regulating mitochondrial autophagy.

### 2.5. The Optimal Active Constituent 30% EF Exerts Anti-PD Action Through Up-Regulated 4E-BP1

The ameliorative effect of the optimal constituent 30% EF against PD induced by MPTP was evaluated using RPPA analysis in the PD cellular model. The principal component analysis (PCA) plot indicated that 6-OHDA samples and 30% EF samples could be isolated well using PCA1 components, with an explanation rate of 44.7% (Figure 6A), indicating that they were clearly distinguished from each other. The volcano plot was used for the visualization of significantly differentially expressed (SDE) proteins, and the expression of 4E-BP1, ROS1, WTAP, and Erk5 up-regulation, SerpinB2 down-regulation (Figure 6B). To elucidate the functions of differentially expressed genes, we conducted GO enrichment analysis. A total of 305 SDE proteins between the 6-OHDA and 30% EF groups were obtained (absolute value of log2 fold change <0.5 and adjusted *p* value < 0.05). SDE proteins were mainly enriched in the following processes in molecular function (MF): phosphatase binding, protein phosphatase binding, and eukaryotic initiation factor 4E binding (Figure 6C). To further investigate the effect of 4E-BP1 on 6-OHDA induced PD, we detected the protein levels of 4E-BP1 and found that the 6-OHDA group significantly decreased its expression, while the 30% EF group brought back to the control level (Figure 6D).

### 2.6. Identification of Main Chemical Components in the Optimal Active Constituent 30% EF Against PD

To identify the main components in 30% EF, we utilized UPLC-Q-Exactive Orbitrap/MS for our analysis. The five predominant components identified were cryptochlorogenic acid, chlorogenic acid, asperuloside, caffeic acid, and asperulosidic acid. Their total ion chromatograms are depicted in Figure 7A. Cryptochlorogenic acid presented a [M-H]-ion at m/z 353.0876, and the corresponding product ions at m/z 263, 191, 179, 173, 135, and 93 (Figure 7B). Caffeic acid generated a [M-H]-ion at m/z 179.0339 and released characteristic fragment ions at m/z 135, and 107 (Figure 7C). Chlorogenic acid showed a [M-H]-ions at m/z 353.0876, and the major fragment ions at m/z 340, 191, 173, 161, 135, 127, 93, and 85 (Figure 7D). Asperulosidic acid gave the [M-H]-ion at m/z 431.1192 and yielded seven diagnostic fragment ions at m/z 251, 165, 121, 101, 89, 71, and 59 (Figure 7E). Asperuloside exhibits a precursor ion [M-H+HCOOH]-at m/z 459.1144 and the prominent product ion at m/z 413, 381, 251, 233, 191, 147, and 119 (Figure 7F). Chlorogenic acid isomer showed a [M-H]-ions at m/z 353.0876, and the major fragment ions at m/z 191, 161, and 85 (Figure 7G). Their retention time, ionization, molecular fragment, precursor ion, and molecular formula are shown in Appendix A. Their structures are shown in Appendix A.

### 2.7. Molecular Docking Simulation, SPR, and Locomotion Assays Further Verify the Anti-PD Effect and 4E-BP1 Associated Mechanism of the Key Active Constituent 30% EF

To explore the interaction between 4E-BP1 (PDB: 2jgb) and the main components (cryptochlorogenic acid, caffeic acid, chlorogenic acid, asperulosidic acid, and asperuloside) in the key active constituent 30% EF, as well as the positive drug rasagiline, we performed molecular docking analyses. The molecular docking process was repeated three times, as detailed in Table 1. Molecular docking visualization images illustrating the favorable interactions of 4E-BP1 with the main active components are presented in Figure 8A–F. A lower binding energy indicates a higher binding affinity, implying that the ligand has a stronger propensity to bind to the receptor components. Cryptochlorogenic acid, caffeic acid, and chlorogenic acid exhibited the lowest binding energies when docked with 4E-BP1, indicating a high degree of binding affinity. Consequently, 4E-BP1 is likely to be a pivotal target in the modulation of the anti-PD action of these components.

To further validate the direct interaction of the main components (cryptochlorogenic acid, caffeic acid, and chlorogenic acid) in the key active constituent 30% EF with 4E-BP1, their binding affinity was estimated by SPR. As shown in Figure 8G–I, the equilibrium dissociation constant (K_D_) values for the interactions between cryptochlorogenic acid and 4E-BP1, caffeic acid and 4E-BP1, as well as chlorogenic acid and 4E-BP1 are 4.469 × 10^−6^ M, 1.292 × 10^−5^ M, and 5.557 × 10^−5^ M, respectively.

In addition, we performed an in vivo assessment to further confirm the alleviative effect of the main components in 30% EF on PD. The exploratory behaviors, as indicated by swimming trajectories, were significantly reduced in the PD model group compared to the control. Conversely, TWE, cryptochlorogenic acid, chlorogenic acid, asperuloside, caffeic acid, and asperulosidic acid reversed this decrease, increasing the time spent and distance traveled (Figure 9A–C). When zebrafish were co-treated with MPTP and active components from 30% EF (cryptochlorogenic acid, chlorogenic acid, asperuloside, caffeic acid, and asperulosidic acid), there was a remarkable increase in the total distance and velocity, indicating that these active components could alleviate MPTP-induced PD-like behavior. Among them, cryptochlorogenic acid demonstrated the most effective alleviation. By comparing the locomotion distance and speed of zebrafish treated with different concentrations (100 μM, 150 μM, 200 μM) of cryptochlorogenic acid, we confirmed its neuroprotective ability (Appendix A).

## 3. Discussion

PD is a chronic neurodegenerative disease marked by diverse motor and non-motor symptoms, profoundly impacting patient quality of life [60]. The key pathological hallmarks of PD include the depletion of dopaminergic neurons in the substantia nigra and striatum, and the accumulation of Lewy bodies composed of synaptic nucleoproteins [61,62]. Novel functional food, has exhibited promising prospects for treating PD, effectively alleviating both the motor and non-motor symptoms of PD. Although EUOL has garnered attention for its therapeutic potential in neurological disorder intervention, including Alzheimer’s disease and epilepsy [33,63], its efficacy, key active constituent, and pharmacological mechanisms against PD remain largely unknown.

We used a single concentration below LC1, aiming to efficiently identify the key active constituent from EUOL against PD. This method facilitated a cost-effective and rapid identification of potential active constituents, guiding subsequent research. Following this discovery, we have established experiments incorporating low, medium, and high concentrations of the 30% EF, enabling a more comprehensive evaluation of the dose–response relationship and strengthening the rigor of our study. In our previous study, we used an MPTP-induced zebrafish PD model to reveal that chlorogenic acid and *sanghuangprous vaninii* extracts have anti-PD activity [24,64]. In this study, we employed the zebrafish PD model and cellular PD model to uncover the key active components of EUOL extract, which show the best potential for treating PD. It has been reported that dopaminergic neuron loss and behavior aberration are prominent characteristics of PD [45,65,66]. Our findings indicated that the 30% EF could ameliorate the dopaminergic neuron loss of zebrafish induced by MPTP. In addition, the zebrafish swimming velocity and distance traveled were significantly suppressed after exposure to MPTP, which is consistent with the previous study [49,67]. Compared to the TWE, the same concentration of 30% EF showed increased locomotor activity. In addition, as a barrier between peripheral blood and the brain, BBB formation and modulation are important for brain homeostasis. They protect the brain from toxic substances and are involved in vascular and neurodegenerative diseases [68,69]. Here, the 30% EF ameliorated the loss of neuronal vasculature and disorganized vasculature induced by MPTP. These findings suggested that 30% EF is the key active constituent for anti-PD activity.

Autophagy eliminates damaged organelles, misfolded proteins, and stress-related products to maintain cellular homeostasis [70,71]. When the autophagy is disrupted, α-syn deposition inhibits its degradation and exacerbates neuronal death [72,73]. In our research, MPTP induced the overexpression of *α-syn*, interfering with the normal autophagy process. Conversely, 30% EF counteracted this effect by reducing α-syn expression. Atg7 is a core protein in the autophagy pathway, involved in the formation of the autophagosome membrane and catalyzing the process of the autophagosome [74,75,76]. After treatment with 30% EF, the increase in *atg7* mRNA expression indicated that it might promote the formation of autophagosomes and enhance the autophagy pathway. Lc3b is a hallmark protein in the autophagy process, participating in the expansion and closure of autophagosomes, as well as the formation of autophagolysosomes [77]. The expression level of Lc3b typically reflects the extent of autophagic activity within cells [78]. The decrease in *lc3b* mRNA expression implied that the formation of autophagosomes was inhibited, which could be due to the 30% EF suppressing excessive autophagy, thus avoiding unnecessary protein degradation. P62 is a protein within the autophagosome that is released from the autophagosome after fusion with the lysosome [79]. When the autophagic activity is inhibited, P62 protein is continuously accumulated [80]. The decreased *p62* mRNA suggested that the release of p62 after fusion of autophagosomes with lysosomes was inhibited, which could be attributed to the 30% EF inhibiting excessive autophagy, thereby preventing unnecessary protein degradation. In summary, following treatment with 30% EF, an up-regulation of *atg7* mRNA was observed concurrently with concomitant down-regulation of *lc3b* and *p62*, suggesting that 30% EF could modulate the autophagy cascade, transitioning it from an overactive to a more balanced state.

RPPA technology, which is adapted from dot blot technology, simultaneously measures protein expression levels in thousands of samples through the arraying of microspots of protein samples on a solid matrix and probing with highly specific antibodies [52]. By further analyzing our RPPA dataset through differential expression analysis between 6-OHDA and 30% EF, we revealed the up-regulation of proteins that are commonly associated with PD pathogenesis, including 4E-BP1. This is a substrate of LRRK2, mutations of which account for the most common genetic form of familial PD [81]. The former research has shown that increasing 4E-BP1 expression or enhancing 4E-BP1 activation can robustly induce the mitochondrial unfolded protein response and may be an effective strategy for treating several neurodegenerative diseases, particularly PD [54,55]. Our Western blot results confirmed the increased expression of 4E-BP1, which was found in RPPA analysis. 4E-BP1 is a critical regulator of translation initiation, functioning to prevent the assembly of the eIF4F complex by blocking the eIF4E-eIF4G interaction [82].

In the zebrafish PD model, MPTP treatment might alter the intracellular environment, including increased oxidative stress and mitochondrial dysfunction, which caused a decrease in the expression of 4E-BP1 protein. The down-regulation of 4E-BP1 might result in increased formation of the eIF4F complex and potentially promote abnormal protein synthesis. It is worth noting that oxidative stress and mitochondrial dysfunction are also closely related to PD [83]. Some studies have shown that activating the Nrf2/Keap1 pathway can alleviate the PD-like symptoms in animal models [84,85]. In addition, mitochondrial dysfunction is another key factor in the pathogenesis of PD. Mitochondria are the main source of cellular energy production, and their dysfunction can lead to insufficient energy supply and increased production of reactive oxygen species (ROS), thereby exacerbating neuronal damage [86]. Research has found that some natural compounds can improve mitochondrial function by regulating mitochondrial autophagy, thus exerting a neuroprotective effect [87,88]. The 30% EF treatment induced the up-regulation of 4E-BP1, which inhibited the formation of the eIF4F complex, thereby suppressing the synthesis of abnormal proteins, including α-syn, a pathological feature of PD (Figure 10). This action helped maintain cellular protein synthesis balance and reduced MPTP-induced damage, thus alleviating PD symptoms.

To further investigate the anti-PD efficacy and mechanism of the optimal active constituent 30% EF, we conducted UPLC-Q-Exactive Orbitrap/MS analysis. It revealed that cryptochlorogenic acid, caffeic acid, chlorogenic acid, asperulosidic acid, and asperuloside were the main components presented in the 30% EF, which were also found in the TWE of the EUOL extract [26,34]. The tight interaction between the main components and 4E-BP1 will assist in understanding the anti-PD efficacy and mechanism of the 30% EF. Thus, we performed molecular docking and SPR. The results indicated that the main components (cryptochlorogenic acid, caffeic acid, and chlorogenic acid) in the 30% EF exhibited relatively stable binding and high affinity with 4E-BP1. We observed that the K_D_ of cryptochlorogenic acid interacted with 4E-BP1 protein was 3.75  ×  10^−5^ M. Notably, a concentration-dependent negative deviation in the binding curve was observed, implying that the binding of cryptochlorogenic acid to 4E-BP1 might result in a reduced SPR signal, in contrast to the usual positive signal changes observed in binding interactions. This effect could be due to conformational changes in the protein or complex upon binding, leading to modifications in the refractive index at the sensor surface. Combining with the results that the main components of the key active constituent 30% EF significantly ameliorated MPTP-induced PD-like symptoms in zebrafish, our findings suggested that it might have a potential therapeutic effect by activating 4E-BP1 in the context of PD.

## 4. Materials and Methods

### 4.1. Reagents and Chemicals

EUOL was purchased from Bozhou City, China. Cryptochlorogenic acid, chlorogenic acid, asperuloside, caffeic acid, and asperulosidic acid were purchased from Desite Bio-Technology Co., Ltd. (Chengdu, China). MPTP and 6-OHDA were purchased from MCE (Monmouth Junction, NJ, USA). Fetal bovine serum and penicillin–streptomycin solution were purchased from Shanghai VivaCell Biosciences Ltd. (Shanghai, China). Paraformaldehyde (PFA) and 1-phenyl 2-thiourea (PTU) were purchased from Sigma (St. Louis, MO, USA). RNA EASY spin Plus RNA Mini Kit was purchased from Aidlab Biotechnologies Co., Ltd. (Beijing, China). CDNA Synthesis SuperMix and NovoStart^®^ SYBR qPCR SuperMix Plus were purchased from Novoprotein (Shanghai, China). BIAcore T200, CM5 (Series Sensor Chip CM5), 200 mM 1-ethyl-3-(3-dimethylaminopropyl) carbodiimide (EDC), and 50 mM N-hydroxysulfosuccinimide (NHS) were purchased from Cytiva (Gloucestershire, UK).

### 4.2. EUOL Extract Preparation

This experiment utilized the hydrothermal extraction method to prepare EUOL total water extract (TWE) powder. The dried EUOL was finely ground to produce 190 g of powder. The powder was then placed in a flask, and 1900 mL of ultrapure water was added to it. The mixture was refluxed using a constant temperature electric heating mantle for 2 cycles, each lasting 3 h. Following reflux, the solution was subjected to centrifugation, after which the clear upper layer was decanted. The solution was subsequently filtered and concentrated using a rotary evaporator to remove all moisture. Following this, the residue was transferred to a vacuum freeze dryer for 36 h until completely freeze-dried, resulting in a total of 66.6 g of EUOL TWE. The yield of EUOL TWE was 35.1%. The EUOL TWE powder was dissolved in 600 mL of pure water and then subjected to elution using D101 macroporous resin. The elution was performed sequentially with pure water, 10% ethanol, 30% ethanol, 50% ethanol, 70% ethanol, and 95% ethanol. The eluants from different water/ethanol ratios were collected and evaporated by rotary evaporation until all the water was removed. Subsequently, these fractions were then placed in a vacuum freeze dryer for 24 h, resulting in water fraction (WF), 10% ethanol fraction (EF), 30% EF, 50% EF, 70% EF, and 95% EF. All extracts were prepared from the same batch of EUOL powder to ensure consistency in the experimental results. A voucher specimen (ID: EUOL-2019-089) has been deposited in the herbarium of the Biology Institute, Qilu University of Technology (Shandong Academy of Sciences), for future reference.

### 4.3. Zebrafish Maintenance and Grouping

#### 4.3.1. Zebrafish Maintenance

Wild-type zebrafish (*AB* strain) and transgenic zebrafish (*slc18a2*: *GFP* and *VEGF*: *GFP*) were raised in zebrafish housing rooms with temperatures generally controlled between 25 °C and 30 °C, under a 14/10 h light/darkness cycle photoperiod. The water is typically from a recirculating system to ensure stable and clean water quality. The dissolved oxygen level in zebrafish medium was between 5 and 8 mg/L, pH value between 7 and 8, electrical conductivity between 500 and 800 μS/cm, and a corresponding salinity range of 0.25–0.50‰. The embryos were then cleaned and preserved in a bathing medium containing 0.5 mg/L methylene blue. Subsequently, the embryos were incubated in a 28.5 °C incubator. Dechorionation was conducted 4 h before the treatment of zebrafish at 1 dpf. All zebrafish lines were used according to the protocol approved by the animal committee of the Biology Institute of Shandong Academy of Sciences, and followed the Use of Laboratory Animals in China guidelines. The approval number for the ethical clearance was SWS20190406, and the study was initiated on 11 October 2019.

#### 4.3.2. MPTP-Induced Zebrafish PD Model

At 24 hpf, the collected zebrafish embryos were dechorionated. Subsequently, these dechorionated embryos were randomly transferred into 6-well cell culture plates. Each well of the plates contained 5 mL of bathing medium and was seeded with 30 zebrafish eggs. In accordance with the previously published studies, the embryos were then exposed to MPTP to establish PD-like conditions [89,90].

#### 4.3.3. Zebrafish Grouping

Zebrafish were randomly divided into 10 groups: Control (Ctl) group, MPTP-treated group (50 μM), positive drug rasagiline-treated group (1 mM), MPTP plus TWE fraction-treated group (50 μg/mL), MPTP plus WF extract-treated group (10 μg/mL), MPTP plus 10% EF-treated group (50 μg/mL), MPTP plus 30% EF-treated group (50 μg/mL), MPTP plus 50% EF-treated group (50 μg/mL), MPTP plus 70% EF-treated group (50 μg/mL), and MPTP plus 95% EF-treated group (2 μg/mL).

Zebrafish were randomly divided into 8 groups: Ctl group, MPTP-treated group, MPTP plus positive drug rasagiline-treated group, MPTP plus cryptochlorogenic acid-treated group (cryptochlorogenic acid, 100 μM, 150 μM, 200 μM), MPTP plus chlorogenic acid-treated group (chlorogenic acid, 150 μM), MPTP plus asperuloside-treated group (asperuloside, 150 μM), MPTP plus caffeic acid-treated group (caffeic acid, 150 μM), and MPTP plus asperulosidic acid-treated group (asperulosidic acid, 150 μM).

Zebrafish were randomly divided into 6 groups: Ctl group, MPTP-treated group, positive drug rasagiline-treated group, 30% EF low concentration group (25 μg/mL), 30% EF medium concentration group (50 μg/mL), and 30% EF high concentration group (100 μg/mL).

### 4.4. Cell Culture

SH-SY5Y cells were cultured in DMEM culture medium, supplemented with 10% fetal bovine serum and 1% penicillin-streptomycin solution at 37 °C under 5% CO_2_.

### 4.5. Locomotion Analysis of Zebrafish

Zebrafish at 5 dpf were transferred to 48-well plates (one larva per well) with E3 medium (1 mL per well). After a 10 min acclimatization period, the locomotor activity of each zebrafish from different experimental groups was recorded using the Zeblab video tracking system (Viewpoint, Lyon, France) for 15 min for the general locomotor activity test. The Zeblab software 4.2 (Viewpoint, Lyon, France) was used to program the light cycle protocol, which consisted of a 10 min initial acclimatization step and 3 cycles of light and dark phases for the light/dark challenge test. Zeblab software (Viewpoint, Lyon, France) was used for the analysis of digital tracks and the average speed of zebrafish.

### 4.6. Assessment of the Morphology of Dopaminergic Neurons in Zebrafish

The transgenic zebrafish (*slc18a2: GFP*) were used in this assay. After treatment, we randomly selected 6 individuals per group and evaluated the changes in dopaminergic neurons using fluorescent microscopy (Zeiss, Jena, Germany). The length of dopaminergic neurons was measured using Image Pro Plus software 6.0 (Media Cybernetics, Bethesda, MD, USA).

### 4.7. Evaluation of the Development of Neuronal Vasculature in Zebrafish

The transgenic zebrafish (*VEGF: GFP*) were used to evaluate the development of neural vasculature. At 4 h post fertilization, 0.003% PTU was added to the E3 medium to inhibit melanin formation. After treating zebrafish for 96 h, 6 zebrafish from each group were randomly selected and observed under a fluorescence field microscope (Zeiss, Germany) to photograph the development of the vasculature in the brain.

### 4.8. Real-Time Quantitative PCR (RT-qPCR) Analysis

Total RNA was extracted from 6 dpf wild-type *AB* zebrafish (*n* = 30) using the RNA EASY spin Plus RNA Mini Kit and then reverse transcribed into cDNA using the NovoScript^®^ Plus All-in-one 1 st Strand cDNA Synthesis SuperMix according to the instructions. RT-qPCR was carried out using NovoStart^®^ SYBR qPCR SuperMix Plus by Light Cycler^®^ Instrument 96 System Roche, Rotkreuz, Switzerland), and then the levels of candidate genes were normalized to *rpl13a.* The Primer sequences are listed in Appendix A.

### 4.9. Whole-Mount In Situ Hybridization in Zebrafish

Zebrafish at 6 dpf (*n* = 10) were fixed for an overnight period in 4% PFA in phosphate-buffered saline (PBS), as previously described [91]. Antisense *α-syn* riboprobes labeled with digoxigenin were synthesized from linearized template DNA by T7 RNA Polymerase using an in vitro transcription system. Fluorescent (Olympus, Tokyo, Japan) microscopy captured representative images.

### 4.10. Immunofluorescence

SH-SY5Y cells were exposed to 1 mM 6-OHDA for 6 h, and then fixed in 4% PFA for 10 min.

After staining with a 1:200 dilution of Alexa Fluor 488-conjugated secondary antibody and DAPI (0.5 μg/mL) for 5 min, the samples were imaged and analyzed using a confocal laser scanning microscope (Olympus, Japan).

### 4.11. RPPA Analysis

The RPPA principle relies on a simple dot blot concept, which detects specific proteins in crude lysates printed as small dots on solid-phase carriers. Briefly, serially diluted protein lysates (protein content ≥ 1.5 μg/μL) were arrayed onto nitrocellulose-coated slides using a Tecan Fluent 480/780. Over 350 protein microarray chips were prepared using a Quanterix 2470 arrayer high-throughput printing apparatus. The microarray chips were then stained with over 300 different antibodies (one antibody per chip). After staining, we used a Huron Tissue Scope LE 120 high-throughput chip to capture image signals, and then MicroVigene was performed to scan, analyze, and quantify the images of stained slides.

### 4.12. Western Blotting Analysis

Cell lysates were centrifuged to obtain supernatants, which were quantified for protein using a BCA kit. Protein samples, each containing 20 μg, were resolved by 12% SDS-PAGE and then transferred to a nitrocellulose membrane. The membrane was blocked with 5% non-fat milk for 2 h at room temperature before being incubated with a 4E-BP1-specific primary antibody (1:1000, Cell Signaling Technology, Danvers, MA, USA) overnight at 4 °C. After washing with PBS, the membrane was then incubated with an HRP-conjugated secondary antibody (1:2000, Sigma, Burlington, MA, USA) for 2 h. β-actin (1:5000, Sigma) served as a loading control. Western blotting analysis is conducted on the same membrane to detect both the internal control β-actin and target protein 4E-BP1 (Appendix A). The blot was analyzed with ImageJ software 1.53. Protein band intensity was normalized against β-actin and presented as a ratio to the control.

### 4.13. Identification and Characterization of Chemical Constituent in EUOL

We reviewed the literature and found the main components of the TWE of EUOL [26,34]. Based on the above information, we conducted a components analysis of the 30% EF. An analytical UPLC-Q-Exactive Orbitrap/MS (Thermo-Fisher Scientific, Waltham, MA, USA) was employed for the analysis. UPLC separation was performed on a Waters Acuity HSS T3-C18 column (2.1 mm × 100 mm) using a gradient mobile phase of 0.1% formic acid in water and acetonitrile. The gradient program was: 0–10 min, 8–15%; 10–25 min, 15–30%; 25–35 min, 30–45%; 35–45 min, 45–65%; 45–55 min, 65–80%; 55–60 min, 80–95%. The injection volume was 2 μL, the flow rate was 0.3 mL/min, and the column temperature was 35 °C. Detection was by ESI in both positive and negative ion modes. The spray voltages were set at 3500V (+)/3500V (−). The ion transfer tube and auxiliary gas heating temperature were set at 320 °C and 350 °C, respectively. The sheath gas was set at 35 arbitrary units and the auxiliary gas at 10 arbitrary units. Xcalibur software (Version 4.1.31.9, ThermoFisher Scientific) was utilized for data processing. The spray voltages were set at 3500V (+)/3500V (−). The ion transfer tube and auxiliary gas heating temperature were set at 320 °C and 350 °C, respectively. The sheath gas was set at 35 arbitrary units and the auxiliary gas at 10 arbitrary units. Xcalibur software (Version 4.1.31.9, ThermoFisher Scientific) was utilized for data processing.

### 4.14. Molecular Docking

The molecular docking analysis was performed using AutoDock Vina with default parameters to investigate potential interactions between EUOL-derived ligand molecules and the binding sites of 4E-BP1 (PDB: 2JGB). The target protein structure was prepared through water molecule removal, addition of polar hydrogens, and assignment of Gasteiger charges. Ligand structures of key active constituents were retrieved from the Protein Data Bank (https://www.rcsb.org). A docking grid centered at coordinates X = 10.190, Y = 10.903, Z = 10.917 with uniform dimensions (40.0 Å in X/Y/Z directions) was established to encompass the binding site. To ensure reliability, three independent docking runs were performed for each ligand, with the lowest-energy conformation selected for subsequent analysis. Binding affinities (kcal/mol) and interaction patterns, including hydrogen bonds and hydrophobic contacts, were evaluated using PyMOL (version 2.5.4) for 3D visualization. Method validation was achieved through redocking of co-crystallized ligands, maintaining a root-mean-square deviation threshold of ≤2.0 Å to confirm docking precision.

### 4.15. SPR Assay

The binding kinetics between 4E-BP1 and its ligands (cryptochlorogenic acid, chlorogenic acid, and caffeic acid) were evaluated using a BIAcore T200 system (Cytiva) based on SPR technology. In the study, a CM5 biosensor chip was activated by injecting a solution of 50 mM NHS and 200 mM EDC for 7 min. The target protein, 4E-BP1, was diluted to 30 μg/mL in 10 mM acetate buffer (pH 4.0) and immobilized on the chip at a flow rate of 10 μL/min for 420 s. Surface blocking was performed with 1 M ethanolamine (pH 8.5), resulting in a protein coupling level of 10,446.3 RU, as quantified by the difference in response units before and after immobilization. Binding kinetics were assessed using a manual 2-fold dilution series with a flow rate of 30 μL/min, 120 s binding time, and 240 s dissociation time. Cryptochlorogenic acid was tested at concentrations of 0, 3.90, 7.81, 15.62, 31.25, 62.5, and 125 μM; chlorogenic acid and caffeic acid were tested at concentrations of 0, 3.90, 7.81, 15.62, 31.25, and 62.5 μM. These ranges were selected based on preliminary solubility assays and prior studies. Data from multiple-cycle experiments were plotted as response units over time and analyzed using BIAcore T200 software 3.2 with a 1:1 Langmuir binding model to derive kinetic constants, including association and dissociation rates, and the equilibrium dissociation constant. Negative controls (activated/blocked channels without protein) and solvent controls (ligand-free buffer) were included to account for nonspecific binding and buffer effects. All experiments were conducted at 25 °C.

### 4.16. Statistical Analysis

Data analysis was performed with GraphPad Prism 7.0, and results are expressed as mean ± SEM. The T-test was used for two-group comparisons, while a one-way ANOVA with Fisher’s LSD or Tukey’s post hoc test was applied to multi-group comparisons. A *p* < 0.05 indicated a significant difference.

## 5. Conclusions

In summary, we unveiled the key active constituent 30% EF from *EUOL in PD treatment* and the underlying mechanisms. 4E-BP1 was considered to be the core target protein. EUOL up-regulated 4E-BP1 to inhibit the synthesis of abnormal proteins, thus relieving PD symptoms. These findings highlight the potential of EUOL as a novel functional food or adjunct therapy for PD, laying a foundation for its future development and application in supporting PD treatment.

## Figures and Tables

**Figure 1 ijms-26-02762-f001:**
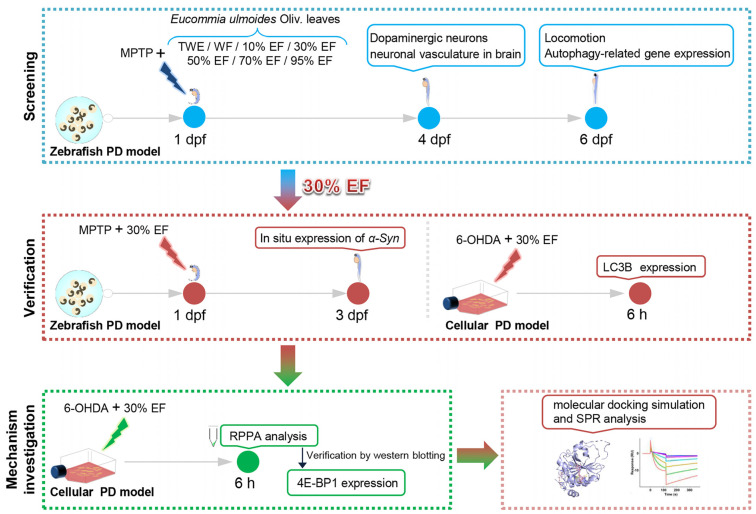
The experimental workflow chart. Zebrafish at 1 day (s) post fertilization (dpf) were exposed to MPTP and different fractions of EUOL extract. The 30% EF was identified to have the best anti-PD activity by evaluating the development of dopaminergic neurons and neuronal vasculature in the brain at 4 dpf. At 6 dpf, we monitored zebrafish behavior and tested the expression of autophagy-related genes. Then, we verified the anti-PD activity of 30% EF using the zebrafish and cellular PD model by detecting the expression of *α-syn* and LC3B. Finally, we investigated the underlying neuroprotective mechanism by locomotion assays, molecular docking simulation, and SPR.

**Figure 2 ijms-26-02762-f002:**
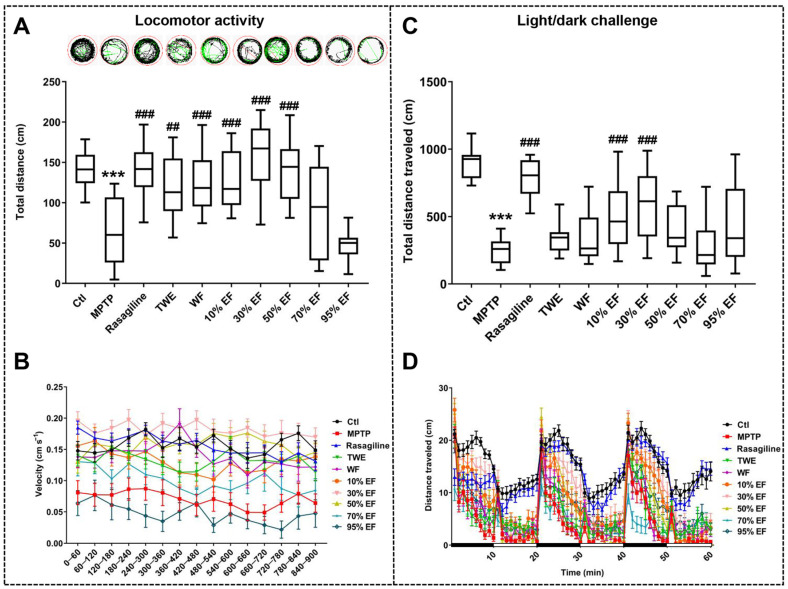
Effect of different fractions of EUOL extract on zebrafish PD-like behavior. (**A**,**C**) Quantitative analysis of total distance traveled by zebrafish (*n* = 24, *** *p* < 0.001 vs. Ctl; ^##^ *p* < 0.01, ^###^ *p* < 0.001 vs. MPTP). (**B**,**D**) Average swimming speed of zebrafish (*n* = 24). The high-speed, medium-speed, and low-speed movement of zebrafish is depicted in red, green, and black lines, respectively.

**Figure 3 ijms-26-02762-f003:**
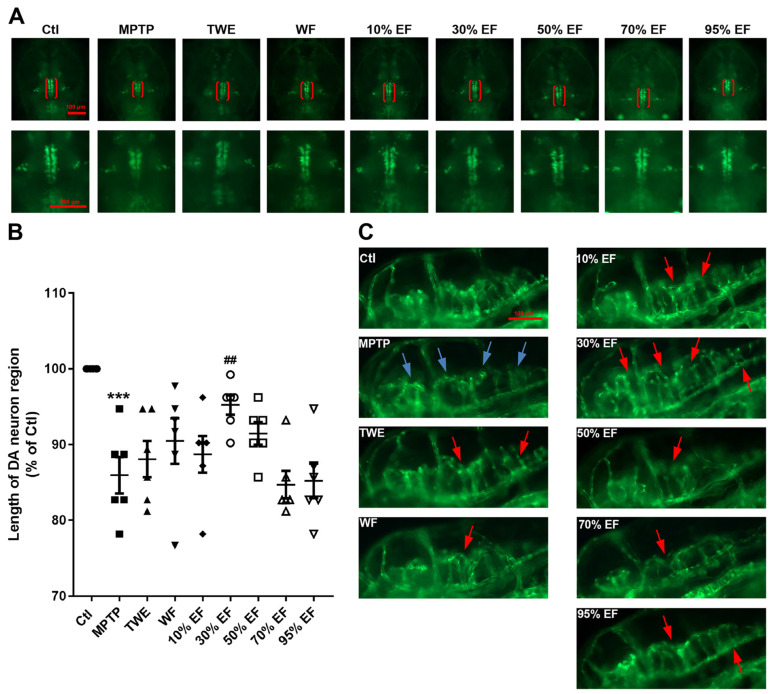
The protective effect of different fractions of EUOL extract on dopaminergic neurons and blood vessels in zebrafish PD model. (**A**) Representative fluorescence microscopy images of dopaminergic neuron regions in zebrafish (red brackets). (**B**) Statistical analysis of the length of the dopaminergic neuron regions of different treated groups (*n* = 12, *** *p* < 0.001 vs. Ctl; ^##^ *p* < 0.01 vs. MPTP). (**C**) Representative fluorescence images of zebrafish in different treated groups. Blue arrows indicate the loss of cerebral vessels, while red arrows indicate cerebral vessels recovery. The scale bar is 100 µm.

**Figure 4 ijms-26-02762-f004:**
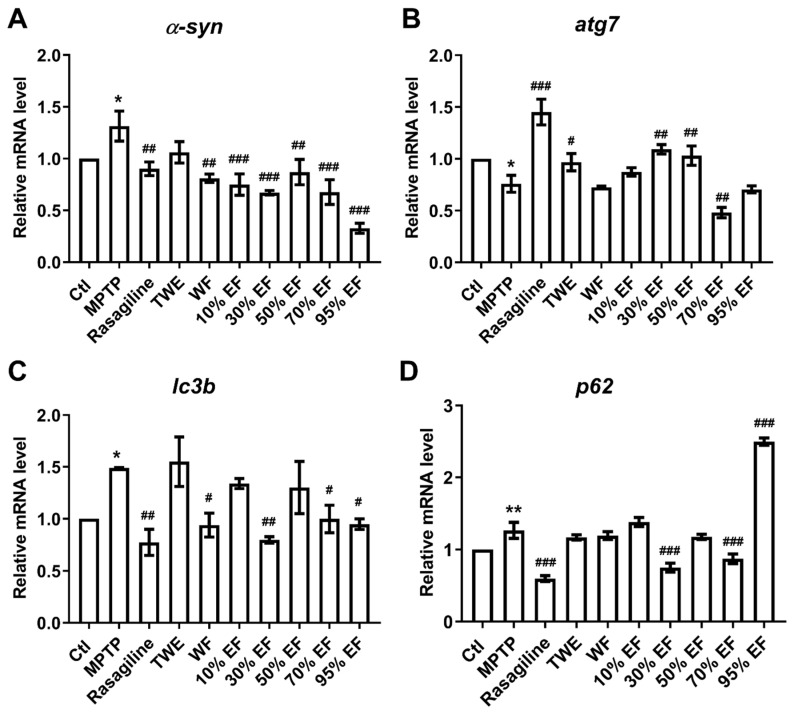
The expression of key genes involved in PD after the treatment with different fractions of EUOL extract. The amount of gene expression ((**A**): *α-syn*, (**B**): *atg7*, (**C**): *lc3b*, and (**D**): *p62*) was exhibited as the relative expression compared with the Ctl. *n*  =  3, * *p*  <  0.05, ** *p*  <  0.01 vs. Ctl; ^#^ *p*  <  0.05, ^##^ *p*  <  0.01, ^###^ *p*  <  0.001 vs. MPTP.

**Figure 5 ijms-26-02762-f005:**
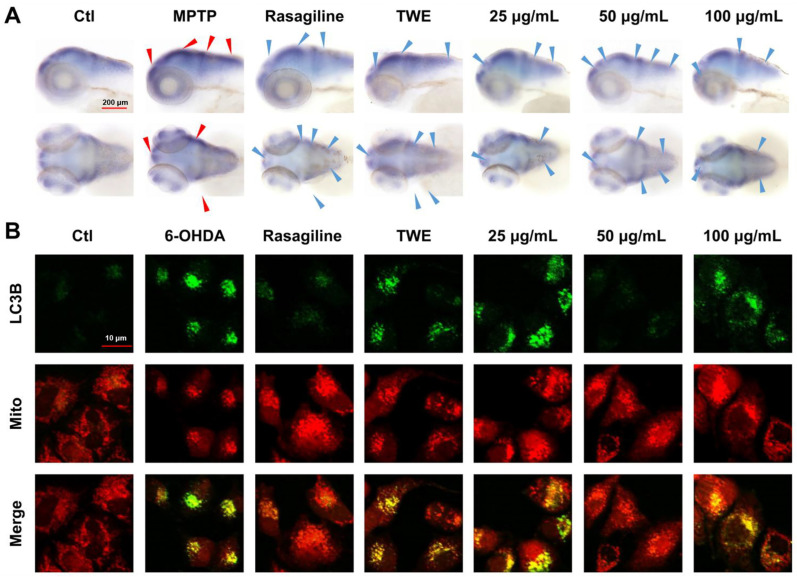
Effect of 30% EF on *α-syn* levels in zebrafish brains and LC3B expression in SH-SY5Y cells. (**A**) The expression of *α-syn* in the CNS of zebrafish larvae after TWE or 30% EF treatment. Red and blue arrows indicate increased and decreased expression of *α-syn* compared to Ctl and MPTP groups, respectively. The scale bar: 200 µm. (**B**) Subcellular distribution of GFP-LC3B and RFP-Mito were visualized on a confocal microscope. Scale bar: 10 μm.

**Figure 6 ijms-26-02762-f006:**
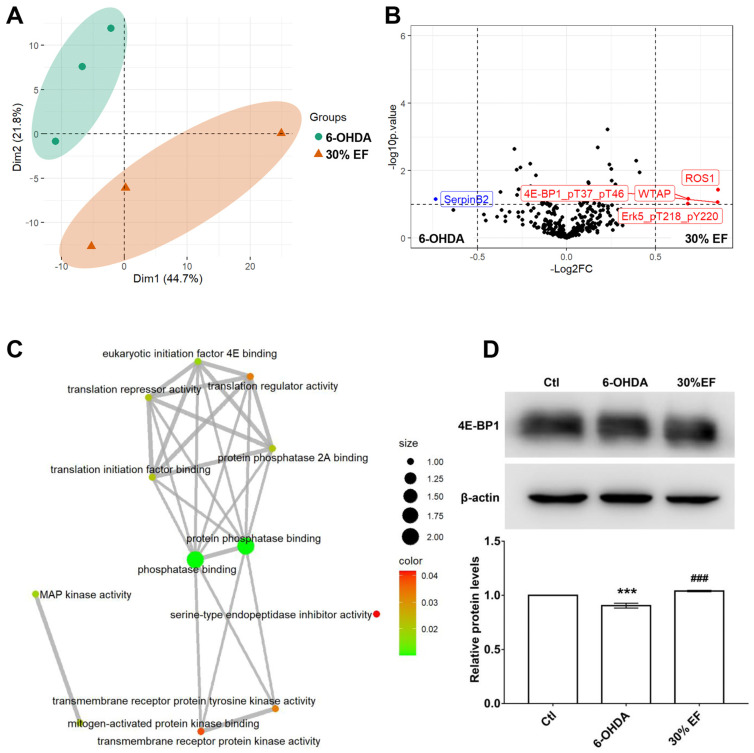
Preliminarily unveiling the underlying mechanism of the 30% EF against PD. (**A**) The PCA of the two datasets. Groups are labeled by different colors (green represents the 6-OHDA group; orange represents the 30% EF group). Microarray platforms are labeled by different shapes (circle represents 6-OHDA group; triangle represents 30% EF group). (**B**) The volcano plot shows differential protein expressions, comparing 6-OHDA vs. Ctl groups. Blue dots denote down-regulated proteins, red dots indicate up-regulated proteins, and black dots signify proteins with non-significant expression changes. (**C**) GO biological function analysis. (**D**) Western blot analysis of 4E-BP1. *** *p* <  0.001 vs. Ctl; ^###^ *p*  <  0.001 vs. 6-OHDA.

**Figure 7 ijms-26-02762-f007:**
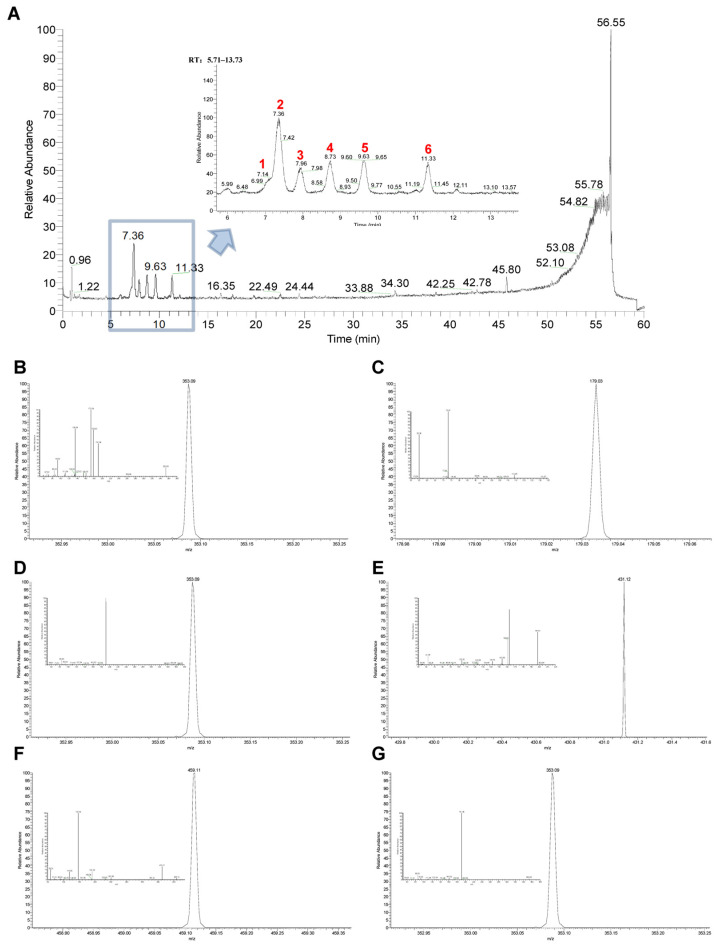
Total ion chromatograms and mass spectrogram of the main components of 30% EF using UPLC-Q-Exactive Orbitrap/MS. (**A**) Total ion chromatograms (peaks 1–6 correspond to the compounds shown in panels (**B**–**G**). (**B**–**G**) Mass spectrum of cryptochlorogenic acid, caffeic acid, chlorogenic acid, asperulosidic acid, asperuloside, and chlorogenic acid isomer in negative ion mode.

**Figure 8 ijms-26-02762-f008:**
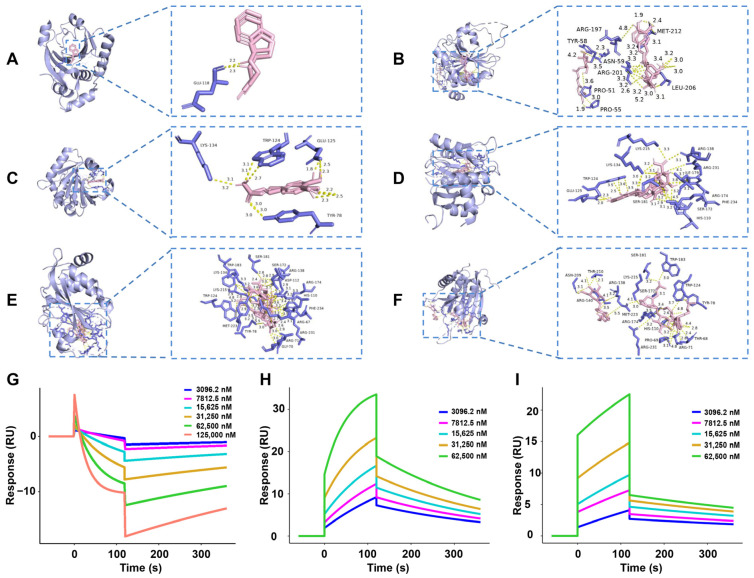
The strong binding ability of the main components in 30% EF with 4E-BP1. (**A**–**F**) The binding interaction of 4E-BP1 with rasagiline, cryptochlorogenic acid, caffeic acid, chlorogenic acid, asperulosidic acid, and asperuloside, respectively. (**G**–**I**) SPR analysis of the binding affinity of cryptochlorogenic acid, caffeic acid, and chlorogenic acid to 4E-BP1, respectively.

**Figure 9 ijms-26-02762-f009:**
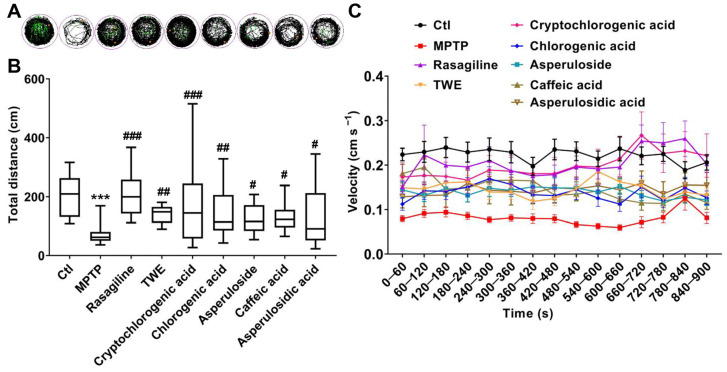
Anti-PD action of the main components in the key active constituent 30% EF by analyzing PD-like behavior. (**A**) The behavioral trajectories of zebrafish under pharmacological intervention. The zebrafish’s high-speed, medium-speed, and low-speed movement is depicted in red, green, and black lines, respectively. (**B**) The total distance traveled by zebrafish with different treatments (*n* = 24, *** *p* < 0.001 vs. Ctl; ^#^ *p*  <  0.05, ^##^ *p*  <  0.01, ^###^ *p*  <  0.001 vs. MPTP). (**C**) Average swimming speed of zebrafish.

**Figure 10 ijms-26-02762-f010:**
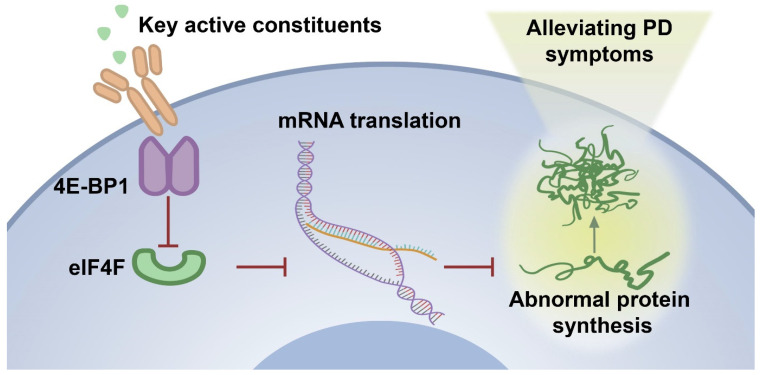
The proposed mechanism underlying the effect of 30% EF on activating 4E-BP1 under PD conditions. MPTP-induced intracellular alterations lead to a decrease in 4E-BP1 expression, enhancing eIF4F complex formation and potentially facilitating abnormal protein synthesis. In contrast, the application of 30% EF counteracts these effects by up-regulating 4E-BP1, which restores translational homeostasis and cellular damage, ultimately alleviating Parkinsonian symptoms.

**Table 1 ijms-26-02762-t001:** Molecular docking results (kcal/mol).

Protein	Ligand	Affinity (kcal/mol)
4E-BP1 (2jgb)	Rasagiline	−4.93 ± 0.12
4E-BP1 (2jgb)	Cryptochlorogenic acid	−9.26 ± 0.76
4E-BP1 (2jgb)	Caffeic acid	−7.98 ± 0.52
4E-BP1 (2jgb)	Chlorogenic acid	−7.79 ± 0.66
4E-BP1 (2jgb)	Asperulosidic acid	−7.71 ± 0.08
4E-BP1 (2jgb)	Asperuloside	−5.38 ± 0.90

## Data Availability

The datasets generated during and/or analyzed during the current study are available from the corresponding author upon reasonable request.

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
