# Peer review of "Identification of Key Active Constituents in Eucommia ulmoides Oliv. Leaves Against Parkinson’s Disease and the Alleviative Effects via 4E-BP1 Up-Regulation"

_ijms, 2025, doi:10.3390/ijms26062762_

Round 1
Reviewer 1 Report
Comments and Suggestions for Authors
The document “Discovery of key active components and mechanisms associated with 4E-BP1 from Eucommia ulmoides Oliv. leaves against Parkinson's disease by integrated multi-model analysis, Phytochemical and reverse phase protein array” is an interesting work with important data. I mention some details:
- They do not mention the amount of plant they used, nor the identification of the same. Present the identification number
- Is the test with zebrafish validated?
- The compounds are listed (1-6) in the chromatogram but not in the description
- Only the fraction with 30% EF was identified. It would be missing from the others that were active
- I consider that they have good results of Molecular Docking
None
Author Response
The document “Discovery of key active components and mechanisms associated with 4E-BP1 from Eucommia ulmoides Oliv. leaves against Parkinson's disease by integrated multi-model analysis, Phytochemical and reverse phase protein array” is an interesting work with important data. I mention some details:
- They do not mention the amount of plant they used, nor the identification of the same. Present the identification number.
Thank you for your valuable suggestion. The dried leaves of Eucommia ulmoides Oliv. (EUOL) were purchased from Bozhou City, China, a region known for its authentic herbal material production, as shown in Section 4.1 (Reagents and chemicals). As detailed in Section 4.2 (EUOL extract preparation), 190 g of dried EUOL powder was subjected to hydrothermal extraction with 1900 mL of ultrapure water, yielding 66.6 g of total water extract (TWE) with a yield of 35.1%. The extraction process and plant authentication comply with Good Laboratory Practice guidelines. The plant material was taxonomically identified following the standards outlined in the Chinese Pharmacopoeia (2020 edition). A voucher specimen (ID: EUOL-2019-089) has been deposited in the herbarium of the Biology Institute, Qilu University of Technology (Shandong Academy of Sciences), for future reference (lines 175-178, highlighted in blue).
- Is the test with zebrafish validated?
We are deeply thankful for your question.The zebrafish PD model used in this study is well-established and has been validated in the previous publications. The increasing and extensive use of the zebrafish PD model in drug screening and mechanism exploration underscores the recognition and reliability of zebrafish as a valuable model organism in PD study [1, 2]. Zebrafish PD models display similar pathological features to PD patients, including the presence of α-syn, the progressive loss of dopamine-producing neurons in the brain, and motor disorders [3, 4]. In addition, with approximately 70% of human genes having zebrafish orthologues, the genetic similarity supports the relevance of findings in this model [5]. Furthermore, the reliability of this model has been demonstrated by showing consistent results across multiple experimental groups and replicates (lines 96-102, highlighted in blue).
- The compounds are listed (1-6) in the chromatogram but not in the description
Thank you for highlighting this issue. We have revised the legend of figure 7 to include the description of the compounds corresponding to peaks 1–6 in the chromatogram (lines 278-279, highlighted in blue).
- Only the fraction with 30% EF was identified. It would be missing from the others that were active
Thank you again for your constructive feedback. The primary objective of this study was to systematically identify the optimal active constituents of EUOL against PD. While other ethanol fractions (e.g., 10% EF, 50% EF) showed partial activity, the 30% EF demonstrated the most significant anti-PD effects across all assays (locomotion, neuronal protection, vascular structural integrity, and autophagy regulation). Thus, we focused on identifying the active compounds from the 30% EF.
- I consider that they have good results of Molecular Docking
We appreciate your positive feedback on our molecular docking results.
References:
- MacRae, C.A. and R.T. Peterson, Zebrafish as tools for drug discovery. Nature Reviews Drug Discovery, 2015. 14(10): p. 721-731.
- Patton, E.E., L.I. Zon, and D.M. Langenau, Zebrafish disease models in drug discovery: from preclinical modelling to clinical trials. Nature Reviews Drug Discovery, 2021. 20(8): p. 611-628.
- Razali, K., et al., The Promise of the Zebrafish Model for Parkinson's Disease: Today's Science and Tomorrow's Treatment. Frontiers in Genetics, 2021. 12.
- Doyle, J.M. and R.P. Croll, A Critical Review of Zebrafish Models of Parkinson's Disease. Frontiers in Pharmacology, 2022. 13.
- Howe, K., et al., The zebrafish reference genome sequence and its relationship to the human genome. Nature, 2013. 496(7446): p. 498-503.
Reviewer 2 Report
Comments and Suggestions for Authors
Using an animal model (zebrafish), the authors investigated the mechanism of the protective and therapeutic effects of Eucommia ulmoides Oliv leaf extract (EUOL) on neurodegenerative damage (Parkinson's disease). The study is a multicenter, multinational and extensive research. Thanks to this collaboration, the researchers could conduct experiments accurately and perform numerous analyses using modern techniques and methods. This would not have been possible in a single laboratory.
The authors proposed a mechanism for the protective effect of Eucommia ulmoides Oliv leaf extract on neurodegenerative changes in Parkinson's disease and a site for therapeutic intervention ( alleviating Parkinsonian symptoms by counteracting abnormal protein synthesis and cell damage by upregulating 4E-BP1 and restoring translational homeostasis).
In addition, the authors determined the optimal concentration and identified the active substances responsible for the protective, restorative effects of Eucommia ulmoides Oliv leaf extract. The manuscript is well-organized and documented. I am impressed with the work the authors put into their study.
Author Response
Thank you for your positive feedback and recognition of our work.
Reviewer 3 Report
Comments and Suggestions for Authors
Article entitled "Unveiling the Key Active Constituents and 4E-BP1 Associated 2 Mechanisms of Eucommia ulmoides Oliv. Leaves Against Park-3 inson's Disease Through Integrated Multi-model, Phytochemi-4 cal, and Reverse Phase Protein Array Analysis" by Yuqing Li et al., found that Eucommia ulmoides Oliv. leaves (EUOL) significantly relieved MPTP-induced lo-28 comotor impairments, increased the length of dopaminergic neurons, inhibited the loss of neuronal 29 vasculature, and regulated the misexpression of autophagy-related genes, suggesting that EUOL could serve as a promising candidate for supporting PD treatment. Article in general is well written however, there are some main concerns before its publication:
Title
I suggest to reduced the title it´s quite big and no very clear
Introduction
I suggest authors to add an image of the Eucommia ulmoides it is always quite usefull for non-expert readers and quite visual.
Please move Figure 1 to methods sections, since it´s confusing on introduction
Material and methods
How do you ensure that EUOL was the correct plant? do you have a voucher number of identification? do you time and date of collection? do you have an specialist from an herbarium?
Do you have a chromatographic profile for the EUOL extract or how do you assure the chemical containce of your extract?
What type of statistics did you use for the molecular docking analysis, please add more information on such.
statistical analysis is not clear for what type of analysis did you use ANOVA for WB, DOcking, SPR assay or which? Please correct this section and add information for each experiment.
Results
Figure 2D is quite fuzzy please improve such image.
Figure 3B is ovelapped please try to improve that figure
Add scale to micrographs otherwise it is difficult to know what are we looking at
Figure 9A its blurred what we see at the top of the graphs please try to modify it so it could be clearer.
Conclusion
Try to add what do you think it could be the possible clinical application (in-case) of your results, for non-expert readers.
Author Response
Article entitled "Unveiling the Key Active Constituents and 4E-BP1 Associated 2 Mechanisms of Eucommia ulmoides Oliv. Leaves Against Park-3 inson's Disease Through Integrated Multi-model, Phytochemi-4 cal, and Reverse Phase Protein Array Analysis" by Yuqing Li et al., found that Eucommia ulmoides Oliv. leaves (EUOL) significantly relieved MPTP-induced lo-28 comotor impairments, increased the length of dopaminergic neurons, inhibited the loss of neuronal 29 vasculature, and regulated the misexpression of autophagy-related genes, suggesting that EUOL could serve as a promising candidate for supporting PD treatment. Article in general is well written however, there are some main concerns before its publication:
1.Title
I suggest to reduce the title it´s quite big and no very clear
Thank you for your suggestion on the title. We have revised it to "Identification of Key Active Constituents in Eucommia ulmoides Oliv. Leaves Against Parkinson’s Disease and the Alleviative Effects via 4E-BP1 Upregulation" (lines 5-8, highlighted in blue).
2.Introduction
2.1 I suggest authors to add an image of the Eucommia ulmoides it is always quite usefull for non-expert readers and quite visual.
We agree with your suggestion and have added an image of Eucommia ulmoides in the Graphical Abstract.
2.2 Please move Figure 1 to methods sections, since it´s confusing on introduction
Thank you for your thoughtful suggestion. While we acknowledge the potential concern, we respectfully retain Figure 1 in the Introduction since it is the experimental workflow. It serves as a visual roadmap for the study design, summarizing the hypothesis-driven workflow to orient readers before methodological details.
- Material and methods
3.1 How do you ensure that EUOL was the correct plant? do you have a voucher number of identification? do you time and date of collection? do you have a specialist from an herbarium?
We appreciate your question. The voucher specimen (ID: EUOL-2019-089) has been deposited in the herbarium of the Biology Institute, Qilu University of Technology (Shandong Academy of Sciences). The plant material was collected from Bozhou City, China, in October 2019 and authenticated by botanists specializing in medicinal plants. This information has now been added to Section 4.2 (lines 470-473, highlighted in blue).
3.2 Do you have a chromatographic profile for the EUOL extract or how do you assure the chemical containce of your extract?
We sincerely thank you for raising this critical question. To ensure the chemical consistency of the EUOL extract, UPLC-Q-Exactive Orbitrap/MS was performed on the 30% EF. The identification of key constituents (e.g., cryptochlorogenic acid, chlorogenic acid, asperuloside, caffeic acid, and asperulosidic acid) was rigorously validated by comparison with reference standards, including retention times, UV spectra, and MS/MS fragmentation patterns. Representative chromatograms and mass spectra, alongside quantitative results, are provided in Figure 7 and Supplementary Table S2, confirming the reproducibility and chemical fidelity of the extract.
3.3 What type of statistics did you use for the molecular docking analysis, please add more information on such.
We sincerely thank you for raising this important question. Binding energy values (kcal/mol) are presented as mean ± SEM across triplicate experiments to account for variability and enhance statistical robustness. Detailed information can be found in Table 1.
3.4 statistical analysis is not clear for what type of analysis did you use ANOVA for WB, Docking, SPR assay or which? Please correct this section and add information for each experiment.
Thank you for your feedback. The T-test was used for two-group comparisons, while a one-way ANOVA with Fisher's LSD or Tukey's post hoc test was applied to multi-group comparisons. P < 0.05 indicated a significant difference (lines 638-640, highlighted in blue).
- Results
4.1 Figure 2D is quite fuzzy please improve such image.
Thank you for pointing out the issue. We have modified the resolution of Figure 2D.
4.2 Figure 3B is ovelapped please try to improve that figure
We sincerely appreciate your comment. We have standardized the data from the Ctl group to 100%, which serves as our baseline for comparison. This normalization was applied to ensure that all sample values are referenced against a common standard, facilitating a direct comparison across different experimental conditions.
4.3 Add scale to micrographs otherwise it is difficult to know what are we looking at
We thank you for bringing this to our attention and have include the scale bar in the micrographs.
4.4 Figure 9A its blurred what we see at the top of the graphs please try to modify it so it could be clearer.
We sincerely appreciate your valuable feedback. We have optimized the resolution of Figure 9A and refined its visual details. This figure illustrates the behavioral trajectories of zebrafish under pharmacological intervention.
5.Conclusion
Try to add what do you think it could be the possible clinical application (in-case) of your results, for non-expert readers.
Thank you for your valuable feedback. We have added the content to our revised manuscript (lines 645-648, highlighted in blue).
Reviewer 4 Report
Comments and Suggestions for Authors
The study presented by Li and Co-workers, effectively combines phytochemical analysis, in vivo zebrafish models, in vitro assays, molecular docking, and protein interaction analysis (SPR), providing a holistic evaluation of Eucommia ulmoides Oliv. Extracts, and combines phytochemical analysis, in vivo zebrafish models, in vitro assays, molecular docking, and protein interaction analysis (SPR), providing a holistic evaluation of Eucommia ulmoides Oliv. extracts. The experimental design involving different ethanol fractions is logical and provides comparative insights into their effectiveness. Additionally, the RPPA and Western blot analysis further validate molecular mechanisms, and the experimental design involving different ethanol fractions is logical and provides comparative insights into their effectiveness. Finally, the RPPA and Western blot analysis further validate molecular mechanisms.
While the manuscript provides valuable insights, there are notable issues regarding clarity, methodological details, and data interpretation. Below are the suggestions:
- While the study is well-structured, the introduction does not clearly state a testable hypothesis. It would benefit from a more direct statement about the expected role of 30% EF in modulating PD-related pathways, in other words, there is a lack of a clear hypothesis.
- Justification of RPPA Analysis is needed, RPPA is primarily used for tumor biomarker analysis. The justification for using it in this neurodegenerative context needs more clarity. Were additional validation steps (e.g., independent proteomics) conducted to confirm key findings?
- Regarding data interpretation, several issues require attention. For instance, while the results indicate that 30% EF significantly affects 4E-BP1 expression, the causal relationship between this effect and neuroprotection is not convincingly established. Additionally, the study primarily attributes neuroprotection to 4E-BP1 modulation; however, other PD-related pathways (e.g., Nrf2/Keap1, mitochondrial function) are not sufficiently discussed. Therefore, it is recommended that the authors expand the mechanistic discussion, particularly addressing how the identified phytochemicals interact with 4E-BP1 to mitigate PD symptoms and whether additional signaling pathways are involved.
- Some methodological details are incomplete. The zebrafish model and behavioral assays lack sufficient descriptions of controls and experimental conditions, and the molecular docking and SPR analysis need more details on ligand selection, scoring functions, and validation methods.
Based on the scientific content and identified issues, minor revisions are recommended before publication.
Author Response
Comments and Suggestions for Authors
The study presented by Li and Co-workers, effectively combines phytochemical analysis, in vivo zebrafish models, in vitro assays, molecular docking, and protein interaction analysis (SPR), providing a holistic evaluation of Eucommia ulmoides Oliv. Extracts, and combines phytochemical analysis, in vivo zebrafish models, in vitro assays, molecular docking, and protein interaction analysis (SPR), providing a holistic evaluation of Eucommia ulmoides Oliv. extracts. The experimental design involving different ethanol fractions is logical and provides comparative insights into their effectiveness. Additionally, the RPPA and Western blot analysis further validate molecular mechanisms, and the experimental design involving different ethanol fractions is logical and provides comparative insights into their effectiveness. Finally, the RPPA and Western blot analysis further validate molecular mechanisms.
While the manuscript provides valuable insights, there are notable issues regarding clarity, methodological details, and data interpretation. Below are the suggestions:
- While the study is well-structured, the introduction does not clearly state a testable hypothesis. It would benefit from a more direct statement about the expected role of 30% EF in modulating PD-related pathways, in other words, there is a lack of a clear hypothesis.
We would like to express our gratitude towards you for this thoughtful suggestion. In the revised manuscript, we have explicitly stated the hypothesis in the Introduction (lines 129-131, highlighted in blue).
- Justification of RPPA Analysis is needed, RPPA is primarily used for tumor biomarker analysis. The justification for using it in this neurodegenerative context needs more clarity. Were additional validation steps (e.g., independent proteomics) conducted to confirm key findings?
Thank you for raising this important question. While RPPA is indeed widely applied in oncology, its utility in neurodegenerative diseases is increasingly emerging. The pathological mechanisms of PD involve numerous alterations in protein expression and function. RPPA enables simultaneous detection of multiple protein expression changes, offering a comprehensive view of PD pathogenesis. To confirm the RPPA findings, we conducted the validation experiments. Western blotting validated the altered expression of 4E-BP1 at the protein level, which is consistent with the RPPA analysis. Furthermore, molecular docking and SPR analysis demonstrated that the main components of the extract exhibit strong binding affinity for 4E-BP1, thereby supporting the RPPA results.
- Regarding data interpretation, several issues require attention. For instance, while the results indicate that 30% EF significantly affects 4E-BP1 expression, the causal relationship between this effect and neuroprotection is not convincingly established. Additionally, the study primarily attributes neuroprotection to 4E-BP1 modulation; however, other PD-related pathways (e.g., Nrf2/Keap1, mitochondrial function) are not sufficiently discussed. Therefore, it is recommended that the authors expand the mechanistic discussion, particularly addressing how the identified phytochemicals interact with 4E-BP1 to mitigate PD symptoms and whether additional signaling pathways are involved.
We appreciate your insightful comments. We acknowledge the importance of establishing a causal relationship between the effect of 30% EF on 4E-BP1 expression and neuroprotection. In our study, we employed multiple experimental approaches to address this, including the observation of neuroprotective effects in both zebrafish and cellular PD models. Additionally, RPPA analysis revealed a significant upregulation of 4E-BP1 in response to 30% EF treatment, which was further confirmed by Western blotting. These findings collectively suggest a strong correlation and provide a reasonable basis for inferring a causal relationship. However, we acknowledge that further experimentation is necessary to definitively prove causality.
We agree that a more detailed discussion of how the identified phytochemicals interact with 4E-BP1 to mitigate PD symptoms would enhance the mechanistic understanding. Molecular docking and SPR analyses showed that 30% EF's main components, especially cryptochlorogenic acid, caffeic acid, and chlorogenic acid, have good binding interactions with 4E-BP1. These interactions likely contribute to the activation of 4E-BP1, which may inhibit the synthesis of abnormal proteins and restore translational homeostasis. In addition, we added the discussion of other PD-related pathways involved in this process. We believe these findings provide a solid foundation for understanding the mechanistic basis of 30% EF’s neuroprotective effects (lines 601-609, highlighted in blue).
- Some methodological details are incomplete. The zebrafish model and behavioral assays lack sufficient descriptions of controls and experimental conditions, and the molecular docking and SPR analysis need more details on ligand selection, scoring functions, and validation methods.
We thank you for highlighting these concerns. We have carefully revised the manuscript and made the necessary changes to address your concerns (lines 490-495, 520-523, 598-614, 616-623, 626-629 and 632-634, highlighted in blue).
.
Round 2
Reviewer 1 Report
Comments and Suggestions for Authors
None